# Catalytic diazene synthesis from sterically hindered amines for deaminative functionalization

Taro Tsuji[1], Isora Fukumoto[1], Takara Hario[1], Mikihiro Hayashi [2], Ayumi Osawa [1], Takashi Ohshima [1] ✉ & Ryo Yazaki [1,3] ✉

Primary amines are highly ubiquitous functional groups found in diverse natural products and building blocks. Despite their widespread application as nucleophiles, the potential for facile deaminative functionalization utilizing primary amines, particularly sterically hindered α-tertiary amines, has remained less explored. Herein, we report catalytic direct synthesis of aliphatic diazenes from sterically hindered α-tertiary amines. The catalytic diazene synthetic method exhibits wide functional group tolerance, allowing for expeditious access to a wide array of hitherto-inaccessible, highly congested diazenes in a short time. Noteworthy is that the present catalytic method enables the synthesis of various hetero-diazenes using distinct α-tertiary amines in just one step for the first time. The catalytic process could also be readily incorporated into Fmoc solid-phase peptide synthesis, enabling the synthesis of elastin-derived diazene, which contains 12 amino acid residues. The catalytic diazene synthetic method enables efficient deaminative transformation of C−N bonds into C−halogen, C−H, C−O, C−S, C−Se, and C−C bonds through carbon-centered radical formation.

Primary aliphatic amines are commonly present in a wide variety of natural molecules, including those with biological activity, and have long served as essential elements in pharmaceutical discovery and molecular design[1-3]. However, despite their natural abundance, these nitrogen-containing functionalities have been significantly under-exploited as versatile synthetic platforms for molecular framework modification[4-7]. The highly nucleophilic nature of amines allows for coupling reactions with electrophiles, but deaminative functionalization remains a significant challenge due to the low heterolytic nucleofugality and high homolytic C−N bond dissociation energy of amines (Fig. 1a)[8-10]. Furthermore, the characteristic basic nature of amines limits the catalytic use of metal reagents, hindering the effective implementation of deaminative cross-coupling reactions.

Several approaches to generate alkyl precursors of primary amines through C−N bond activation have been investigated

(Fig. 1b)[11-13]. Among the widely adopted methods is a system utilizing amine-derived Katritzky salts[14-17]. The broad popularity of Katritzky salts stems from the ease of their synthesis and accessible reduction potential, enabling various transition metal-catalyzed and photoredox-catalyzed deaminative transformations. However, Katritzky salts cannot be synthesized from sterically hindered α-tertiary amines, which limits their applicability. New strategies that differ from conventional deamination reactions involve the combination of Schiff bases and Ir photoredox catalysts[18-21], or isodiazene from anomeric amide[22,23]. The method using Schiff bases achieved deaminative functionalization of α-tertiary amines through one-step preactivation. The strategy of isodiazene efficiently functionalized the α-primary and secondary amines using anomeric amide[24]. Despite intensive efforts to develop efficient deaminative functionalization reactions, challenges remain, such as limitations in

[1]Graduate School of Pharmaceutical Sciences, Kyushu University, Maidashi, Higashi-ku, Fukuoka, Japan. [2]Department of Life Science and Applied Chemistry, Graduate School of Engineering, Nagoya Institute of Technology, Gokiso-cho Showa-ku Nagoya-city, Aichi, Japan. [3]Institute for Advanced Study, Kyushu University, Motooka, Nishi-ku, Fukuoka, Japan. ✉e-mail: ohshima@phar.kyushu-u.ac.jp; yazaki@phar.kyushu-u.ac.jp

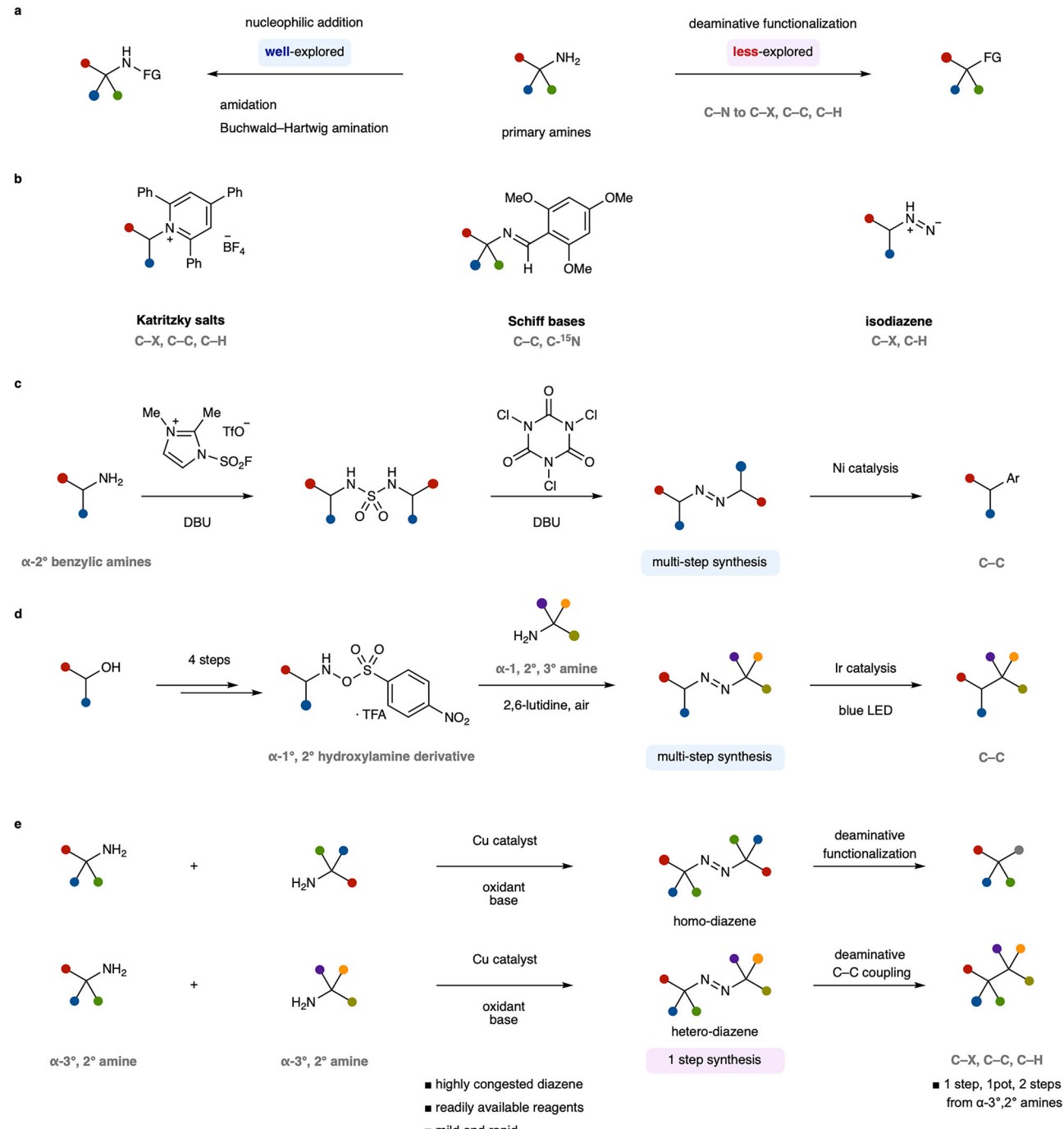

**Fig. 1 | Deaminative functionalization of primary amine. a** Utility of primary amine for conventional nucleophilic amination and deaminative functionalization. **b** State-of-the-art deaminative functionalization of primary amine. **c** Multi-step diazene synthesis for deaminative functionalization via sulfonamide. **d** Multi-step diazene synthesis for deaminative functionalization using hydroxylamine derivatives. **e** This work: catalytic synthesis of diazenes from α-tertiary amines for deaminative functionalization.

functional group transformation and the requirement for stoichiometric amounts of metal activators[25–28].

To overcome these limitations, we focused on sterically minimized aliphatic diazenes, which are typically synthesized from ketones and hydrazines through multi-step processes[29–32]. Although aliphatic diazenes can generate alkyl radicals by releasing nitrogen gas under heated or light-irradiation conditions, commonly serving as a radical initiator in polymer synthesis, investigations of their application in deaminative functionalization are highly limited. Recently reported synthetic methods for deaminative C–C coupling reactions of diazenes

still require multi-step sequences for the preparation of diazenes, as exemplified by sulfonamides (Fig. 1c), and hydroxylamine derivatives (Fig. 1d)[33–35]. Additionally, the range of applicable primary amines is primarily restricted to α-secondary amines. Direct synthesis of aliphatic diazenes from primary aliphatic amines remains scarce compared to the numerous methods available for synthesizing aromatic diazenes. These methods require hazardous reagents[36–40], cumbersome multi-step preparations[33,35,41–47], and have a limited substrate scope[48–50], thus, no catalytic methods have been successfully developed.

**Table 1 | Effect of reaction parameters on the copper-catalyzed diazene synthesis using α-tertiary amine**

| Entry | Variation from the standard conditions | Yield (%) |
|---|---|---|
| 1 | None | >99 |
| 2 | No CuOAc | 7 |
| 3 | Cu(OAc)$_2$ instead of CuOAc | 22 |
| 4 | Fe(OAc)$_2$ instead of CuOAc | 21 |
| 5 | AgOAc instead of CuOAc | 35 |
| 6 | O$_2$ (1 atm) instead of DBDMH | 0 |
| 7 | Oxone (3.0 equiv) instead of DBDMH | 0 |
| 8 | K$_2$S$_2$O$_8$ (3.0 equiv) instead of DBDMH | 0 |
| 9 | TBHP (3.0 equiv) instead of DBDMH | 0 |
| 10 | DCDMH instead of DBDMH | 33 |
| 11 | DIDMH instead of DBDMH | 4 |
| 12 | NBS (3.0 equiv) instead of DBDMH | >99 |
| 13 | K$_2$CO$_3$ instead of DBU | 30 |
| 14 | KO$^t$Bu instead of DBU | 25 |
| 15 | NEt$_3$ instead of DBU | 5 |
| 16 | DABCO instead of DBU | 60 |

Reactions were performed on a 0.20 mmol scale.

Yields were determined by $^1$H NMR spectroscopic analysis.

Here, we develop a catalytic direct synthesis of aliphatic diazenes from sterically hindered amines under mild and rapid conditions (Fig. 1e). The present catalytic method enables the transformation of C–N bonds into C–halogen, C–H, C–O, C–S, C–Se, and C–C (Csp$^3$, Csp$^2$, Csp) bonds through a deaminative carbon-centered radical formation utilizing diazenes with a broad substrate scope.

## Results

We first investigated solvents and reaction times using α-tertiary amine **1a** in the presence of a copper catalyst (Table 1). The desired product **2a** was obtained in a remarkably short reaction time, only 1 min in dimethylformamide (DMF) using readily available CuOAc, 1,3-dibromo-5,5-dimethylhydantoin (DBDMH), and 1,8-diazabicyclo[5.4.0]undec-7-ene (DBU) (entry 1). Acetonitrile (MeCN) also delivered **2a** with the same efficiency as DMF. Without the catalyst, only 7% of **2a** was observed (entry 2). Various catalysts, including Cu(OAc)$_2$, Fe(OAc)$_2$, and AgOAc, exhibited low catalytic performance (entries 3–5). Various oxidants were evaluated next. Oxygen gas and peroxides did not afford **2a** (entries 6–9). In contrast, the use of halogenated oxidants, 1,3-dichloro-5,5-dimethylhydantoin (DCDMH), and 1,3-diiodo-5,5-dimethylhydantoin (DIDMH), afforded **2a**, albeit with low chemical yields (entries 10 and 11). *N*-Bromosuccinimide (NBS) exhibited the same efficiency as DBDMH (entry 12). Base screening revealed that DBU was crucial for efficient reaction progress. Both potassium carbonate and potassium *tert*-butoxide provided **2a** with low efficiency (entries 13 and 14). Although triethylamine delivered **2a** in low chemical yield (entry 15), a

moderate chemical yield was afforded using 1,4-diazabicyclo[2.2.2]octane (DABCO) (entry 16).

Having developed an efficient diazene synthetic method, we next explored the substrate scope using various α-tertiary amines at a 10-min reaction time (Fig. 2). The catalyst loading and equivalents of reagents could be reduced without detrimental effects (5 mol% CuOAc; DBDMH, 1.2 equiv; DBU, 1.0 equiv), and **2a** was isolated in 93% yield. The reaction was successful on a gram scale. A readily available amine hydrochloride salt was also applicable with slight modification of the reaction conditions (**2b**). Substrates with a 5- or 6-membered alkyl ring at the α-position also afforded the product in high yields (**2c, 2d**). Cyano and Weinreb amide functionalities were well tolerated, delivering the products with high yields (**2e–2g**). An amine tri-fluoroacetic acid (TFA) salt having ketone functionality afforded product **2h** in moderate yield. The highly congested adamantyl group was applicable to the present catalysis, although the chemical yield was only moderate (**2i**). Various functional groups, including Boc-protected tertiary amine, cyclic ether, sulfonyl group, and an indanyl group, were also incorporated without any problem (**2j–2m**). It is noteworthy that a sterically hindered unnatural amino acid having a 7-membered alkyl ring at the α-position successfully afforded product **2n** in synthetically useful yield[51,52]. β-Amino acids, having no activating group at the α-position, were applicable, delivering product **2o** in 66% yield. The reaction proceeded smoothly even when 1-adamantylamine, which has no functional group at all, was used (**2p**), indicating that an electron-withdrawing group was not crucial for the present catalysis. Amino alcohol derivatives were also transformed into the diazenes **2q** and **2r**. A variety of dipeptides, such as glycine, alanine, phenylalanine, valine, and asparagine, were used, and the products were obtained in high yields (**2s–2w**). α-Secondary amines are difficult to use as sub-strates under oxidative conditions due to undesired imine formation. However, α-secondary amines were also applicable under slightly modified reaction conditions, resulting in diazenes being isolated in synthetically useful yields (**2x–2z**). Although increasing the amount of copper catalyst effectively suppressed the formation of the imine byproduct, the observation of imine formation even under stoichio-metric copper conditions indicates that copper-mediated pathways may still contribute to imine formation.

The present catalysis was not confined to homo-diazene synthesis. We further investigated the substrate scope involving hetero-coupling with two distinct α-tertiary amines. Hetero-diazene synthesis from distinct amines has been regarded as highly challenging[34,35]. In particular, direct hetero-diazene synthesis using sterically hindered α-tertiary amines has never been achieved, even under stoichiometric conditions. When 3.0 equivalents (0.60 mmol) of the α-tertiary amine **1b** were used, the hetero-diazene **3ab** was isolated in 71% yield (0.15 mmol), which is nearly consistent with the statistical theoretical maximum, along with the concomitant formation of homo-diazenes (**2a**: 0.028 mmol, **2b**: 0.23 mmol). The overall material balance was satisfactory, confirming that the reaction cleanly proceeded. The use of 2.0 equivalents (0.40 mmol) of **1b** also delivered hetero-diazene **3ab** in 66% yield, consistent with the statistical ratio. These results indicate that, although an excess amount of amine is required, the present catalysis is highly effective for the direct synthesis of hetero-diazenes from different α-tertiary amines. α-Tertiary amine without electron-withdrawing groups could be incorporated into the hetero-diazene without any detrimental effects (**3ap, 3dq**). Dipeptides could also be incorporated into hetero-diazene, providing the peptide-amino acid-derived diazenes **3sb** and **3tb** in synthetically useful yields. A diverse range of hetero-diazenes could be synthesized using various combinations of α-tertiary amines (**3cp, 3dp, 3kp, 3dk, 3jd**). Cross-coupling of α-tertiary amine with α-secondary amine successfully provided the hetero-diazene **3py**, although the chemical yield was modest, and a small amount of impurity remained. The wide substrate scope evident in the present catalysis clearly indicates the potential to offer a range of

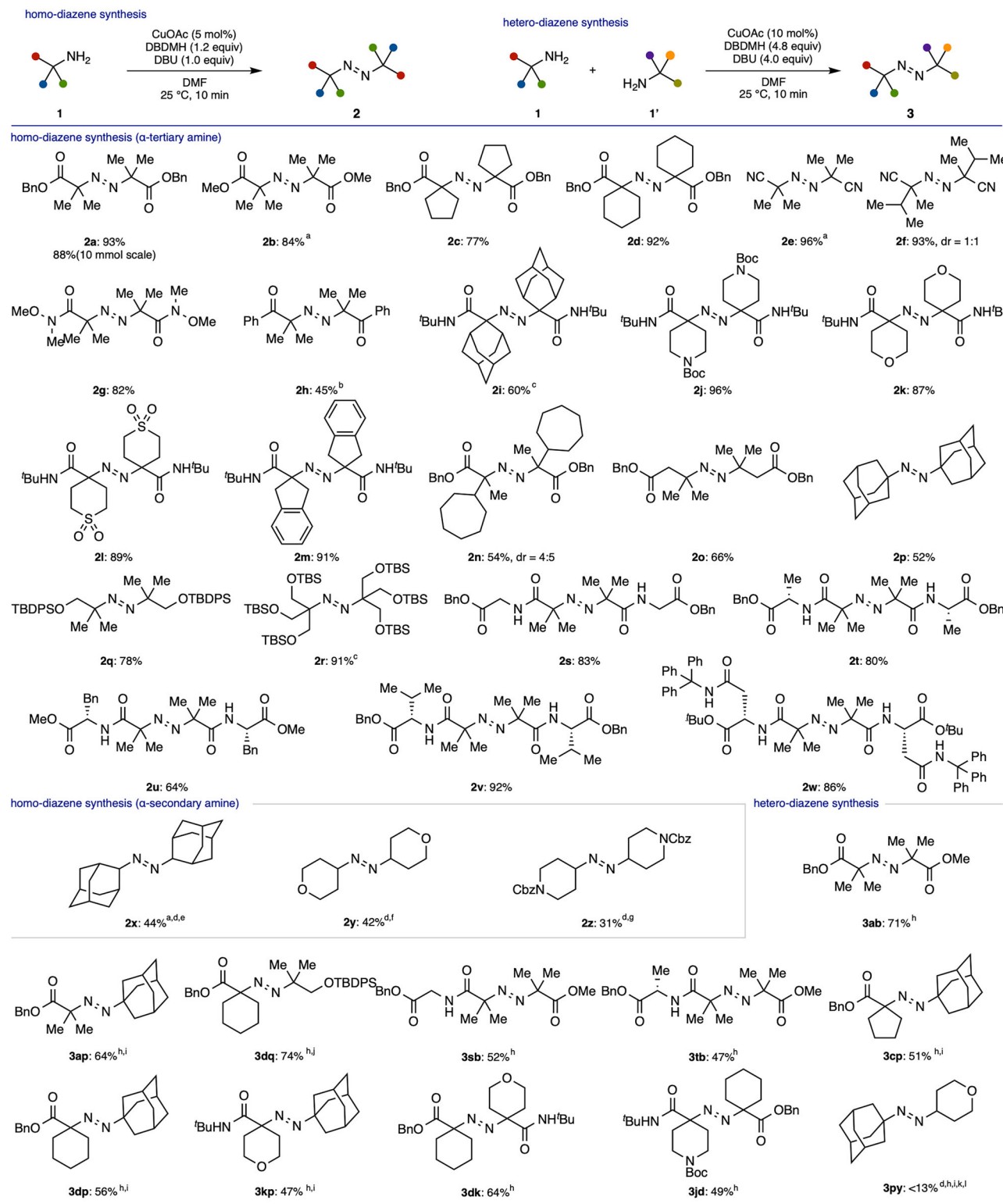

**Fig. 2 | Copper-catalyzed diazene synthesis using α-tertiary and secondary amines: Reactions were conducted on 0.20 mmol scale using 0.2 ml of DMF.** Isolated yields are shown. Diastereomeric ratios were determined by [1]H NMR analysis of crude mixture. [a]**1**•HCl salt (1.0 equiv), DBU (2.0 equiv). [b]**1**•TFA salt (1.0 equiv), DBU (2.0 equiv). [c]DMF (0.4 ml), 3 h. [d]DCDMH was used instead of DBDMH.

[e]CuOAc (20 mol%). [f]CuOAc (50 mol%). [g]CuOAc (100 mol%). 1.0 mmol scale. [h]Three equivalents of the amine on the right side were used. 0.4 ml of DMF. [i]Reaction time was 1 h. [j]0.50 mmol scale. [k]CuOAc (100 mol%). [l]A small amount of impurity was still present after silica gel column chromatography. Boc tert-butoxycarbonyl, Cbz benzoxycarbonyl TBDPS tert-butyldiphenylsilyl, TBS tert-butyldimethylsilyl.

hitherto inaccessible highly congested hetero-diazenes in just one step.

We investigated thermal stability using thermogravimetric analysis (TGA) for several representative compounds, namely **2a, 2r, 2t,**

**2y**, and **3ap**, which possess different hindered structures (Fig. S6). The thermal stability of these compounds was comparable to, or significantly greater than, that of a typical diazene compound (i.e., azobisisobutyronitrile, AIBN), which shows an onset of thermal

**Fig. 3 | Catalytic diazene synthesis in Fmoc solid-phase peptide synthesis.** Catalytic diazene synthesis applied to solid-phase peptide synthesis.

decomposition at approximately 100–125 °C[53]. Therefore, the synthesis reported here may enhance the availability of thermally stable diazene compounds, which could also be valuable in the field of polymer science, as these compounds are commonly used for radical polymerization.

The rapid and broad substrate scope observed in the present catalysis prompted us to further investigate diazene synthesis using larger functionalized molecules. To assess the applicability of the catalysis to peptide-derived diazene synthesis, we applied our method to Fmoc solid-phase peptide synthesis using elastin, a fibrous protein primarily responsible for linking collagen molecules (Fig. 3). The repetitive sequence peptide of elastin was synthesized through Fmoc solid-phase peptide synthesis. Subsequently, peptide **1α** attached to resin was subjected to the optimal reaction conditions, followed by resin cleavage using a TFA cocktail. Peptide-derived diazene **2α** was isolated in 11% yield following 15 steps in Fmoc solid-phase peptide synthesis, as a pure form after HPLC purification, showcasing the high applicability of the present method in the solid-phase system.

Finally, the present catalysis was applied to deaminative functionalization (Fig. 4). Direct transformation of α-tertiary amine into tertiary alkyl halides was successfully achieved for the first time. Under catalytic conditions for synthesizing diazenes, a simple temperature increase from room temperature to 70 °C enabled a remarkable one-step deaminative bromination **4**. Deaminative chlorination from **5** was also achieved using DCDMH instead of DBDMH, clearly demonstrating that the present protocol allows for the use of α-tertiary amines as an alkyl source[4,5,51,54]. The direct deaminative bromination reaction also proceeded using **1d** and **1p**, although the yields were moderate (**6, 7**).

Further utility of the present deaminative functionalization strategy was demonstrated by one-pot transformations. Deaminative iodination was accomplished by synthesizing the diazene followed by adding NaI to the crude mixture (**8**). The addition of triisopropylsilanethiol under a similar protocol enabled one-pot removal of the α-tertiary amine (**9**). The addition of TBHP achieved deaminative hydroxylation, delivering tertiary alcohol **10** in synthetically useful yield. Diazene **2a** also served as 2 equivalents of a tertiary carbon donor for versatile radical coupling reactions. Treatment of iodine under heated conditions provided iodinated product **8** in high yield[55]. Deaminative peroxidation also proceeded in the presence of TBHP (**11**). The introduction of TEMPO was achieved under blue LED light irradiation (**12**). The use of sulfonyl reagents enabled deaminative thioetherification and selenylation (**13, 17**, respectively). Diazene **2a** turned out to be a suitable substrate for C–C coupling reactions, including Csp[3], Csp[2], and Csp. Coupling reactions with silyl enolate provided alkylated product **14** in the presence of a copper catalyst[56]. The NHC-

catalyzed method delivered carbonylated compound **15** in high yield[32]. Alkynylated product **16** was obtained through treatment with a hypervalent iodine reagent.

Hetero-diazenes, which could be directly synthesized in one step from different amines using our catalytic method, were applicable to radical coupling reactions involving denitrogenating. Under blue light irradiation, hetero-diazenes constructed highly congested contiguous quaternary carbon centers, which are generally challenging to assemble, through solvent-cage directed radical cross-coupling (**18-21**)[57], demonstrating the utility of our catalytic diazene synthesis method.

Next, we performed a series of control experiments (Fig. 5). Treatment with DBDMH in the absence of CuOAc and DBU exclusively afforded the *N*-monobromo and dibromo derivatives (**22, 23**) without forming **2a** (Fig. 5a). The addition of CuOAc and DBU to the mixture of **22** and **23** delivered **2a** in quantitative yield, indicating that **22** and **23** serve as intermediates for the generation of diazene **2a**. Omission of CuOAc delivered **2a** in 4% yield (Fig. 5b). Without DBU, a low chemical yield of **2a** was also observed (Fig. 5c). In the course of investigating bases, DABCO, known to form complexes with NBS or NIS through halogen interactions confirmed by X-ray crystallography[58–61], exhibited high performance for generating **2a** (Table 1, entry 16), suggesting that DBU would weaken the N–Br bond or stabilize the bromine radical[62–68].

To further investigate the formation of the aminyl radical, we performed electron spin resonance (EPR) measurements using 5,5-dimethyl-1-pyrroline *N*-oxide (DMPO) as a spin-trapping agent (Fig. 5d). When the reaction was conducted under the standard conditions using NBS as the oxidant in the presence of DMPO, HRMS peak corresponding to the DMPO adduct **24** was observed. Furthermore, EPR measurements successfully detected the signal of the DMPO adduct **24**, suggesting that an aminyl radical species would be generated under the present reaction conditions.

On the basis of a series of control experiments and DFT calculation, the proposed catalytic cycle is shown in Fig. 5e. Without a copper catalyst, the N–N bond formation would proceed through a two-electron mechanism, which was sluggish due to the significant steric hindrance by the two α-tertiary amines. In the presence of a copper catalyst, the reaction rate was significantly enhanced, indicating a potentially distinct reaction pathway compared to the reaction without a copper catalyst. Although the formation of a copper(III) species via oxidative addition of the copper catalyst to intermediate **A** is feasible, DFT calculations suggest that the subsequent generation of aminyl radical species **C** is energetically unfavorable. In contrast, the halogen atom abstraction pathway is more likely and is considered to be the predominant route leading to aminyl radical formation, as supported by EPR analysis. Furthermore,

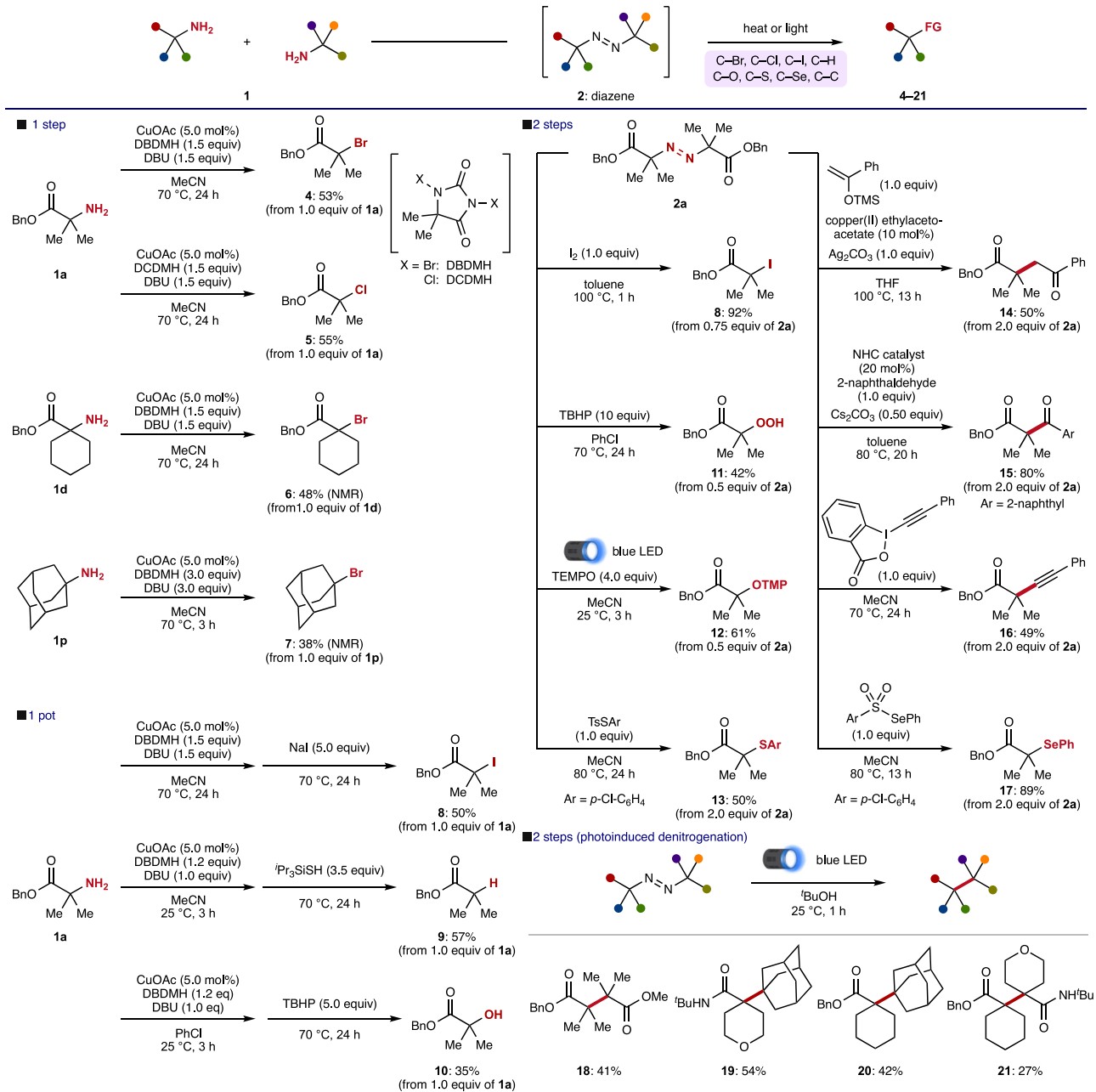

**Fig. 4 | Deaminative functionalization through catalytic diazene synthesis.** Various deaminative functionalizations were performed, including halogenation, hydrogenation, hydroxylation, thiolation, selenation, and C–C bond formation.

the reaction pathway involving aminyl radical species **C** followed by the N–N bond formation to afford intermediate **E**, which is less impacted by steric hindrance, would be operative (calculated $\Delta G^{\ddagger} = 8.8$ kcal/mol, $\Delta G = -38.8$ kcal/mol. See Supplementary Information section 9)[69–76]. The formed halogenated hydrazine **E** would subsequently be converted into the diazene product **2** by DBU without the involvement of copper species[77].

In summary, we developed a catalytic deaminative protocol through direct synthesis of aliphatic diazenes from sterically hindered α-tertiary amines under mild and rapid conditions. A variety of α-tertiary amines could be incorporated into the corresponding diazenes[78], not only homo-diazene but also hetero-diazenes, delivering highly congested diazenes that had not been previously synthesized. The exceptional catalytic efficiency allowed for application to Fmoc solid-phase peptide synthesis, affording elastin-derived diazene. The catalytic diazene synthetic method enabled efficient

transformation of C–N bonds into C–halogen, C–H, C–O, C–S, C–Se, and C–C bonds through carbon-centered radical formation. We believe that these results demonstrate the high potential of the present catalytic diazene synthetic method using sterically hindered amine as an alkyl source.

## Methods

### General procedure for catalytic homo-diazene synthesis

To a 4-ml vial equipped with a magnetic stir bar, CuOAc (1.2 mg, 10 µmol, 5.0 mol%) was added in a glovebox followed by the addition of cold DMF (0.20 ml, 1.0 M), 1,3-dibromo-5,5-dimethylhydantoin (68.6 mg, 0.24 mmol, 1.2 equiv), amine (0.20 mmol, 1.0 equiv), and 1,8-diazabicyclo[5.4.0]undec-7-ene (30 µl, 0.20 mmol, 1.0 equiv) under Ar atmosphere. The reaction mixture was stirred at 25 °C for 10 min and diluted with CH₂Cl₂. The diluted solution was filtered through a silica short column and washed with EtOAc. After evaporation of the organic

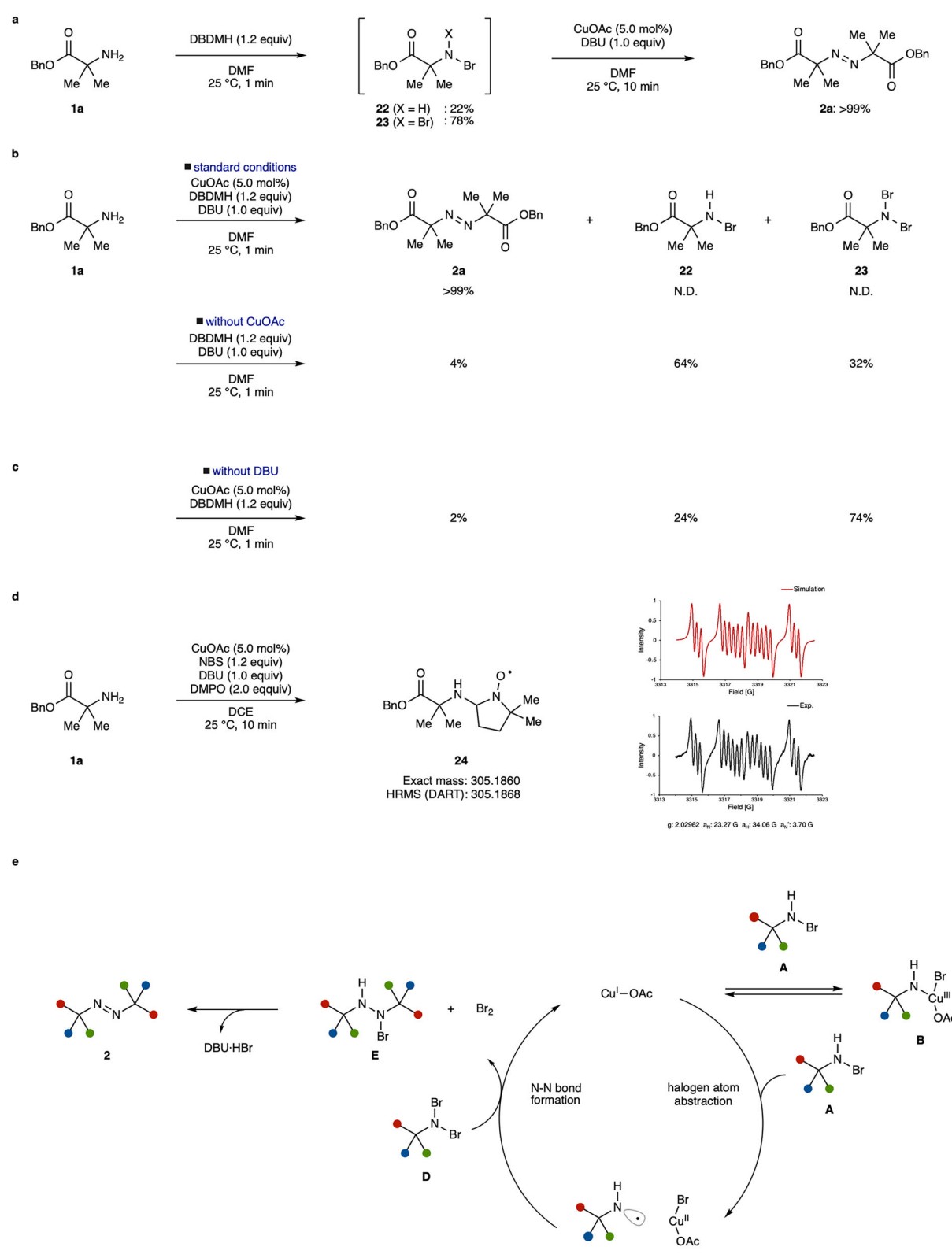

**Fig. 5 | Series of mechanistic studies and proposed catalytic cycle.**
**a** Intermediate formations of **17** and **18** with DBDMH. **b** Control experiments without copper suggested that a copper catalyst significantly facilitated the formation of **2a**. **c** Control experiments without DBU indicated that DBU was crucial for efficient reaction progress. **d** EPR analysis of DMPO adduct **24**. **e** Proposed catalytic cycle. The halogen atom abstraction would first proceed with intermediate **A** to generate copper (II) species and aminyl radical species **C**. Copper(III) species **B** formation would be feasible, but aminyl radical generation would be unfavorable. The N–N bond formation proceeds through aminyl radical species **C** with **D** to afford intermediate **E**.

solvent under reduced pressure, the resultant mixture was purified by silica gel flash chromatography to obtain the desired compound.

## Data availability

The data supporting the findings of this study are available within the article and its Supplementary Information. Data supporting the findings of this manuscript are also available from the corresponding author upon request.

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

## Acknowledgements

This work was financially supported by the JST-FOREST program (JPMJFR2229), KAKENHI grant numbers JP24H01777 for the Grant-in-Aid for Transformative Research Areas (A) Latent Chemical Space from MEXT, JP21A204, JP21H05207, and JP21H05208 for the Grant-in-Aid for Transformative Research Areas (A) Digitalization-driven Transformative Organic Synthesis (Digi-TOS) from MEXT, a Grant-in-Aid for Scientific Research (B) (JP21H02607), a Grant-in-Aid for Basis for Supporting Innovative Drug Discovery and Life Science Research (BINDS) (JP20am0101091, JP22ama121031) and AMED (JP21ak0101167,

JP22ak0101167, JP23ak0101167). T.T. thanks the JSPS for a predoctoral fellowship. T.O. thanks NAGASE Science Technology Foundation for financial support. R.Y. thanks the Takeda Science Foundation, the Noguchi Institute. and the Nakatani Foundation, and the Japan Association for Chemical Innovation, for financial support. We are grateful to Prof. Ken Yamazaki (Okayama University) for fruitful discussion about DFT calculation, Prof Kenichi Yamada, Dr. Masami Abe, and Mr. Eisho Kozakura for EPR measurement.

## Author contributions

T.T. and R.Y. conceived the work, analyzed the data, discussed the results, and all authors co-wrote the manuscript. T.O. checked the manuscript and Supplementary Information. T.T. designed and carried out the experiments with I.F. and T.H. T.T. and A.O. performed DFT calculation. M.H. performed thermogravimetric analysis (TGA).

## Competing interests

The authors declare no competing interests.
