## [Transparent Peer Review file · Nature Communications]

Catalytic Diazene Synthesis from Sterically Hindered Amines for Deaminative Functionalization

Corresponding Author: Dr Ryo Yazaki

Version 0:

Reviewer comments:

Reviewer #1

(Remarks to the Author)

In this manuscript, Yazaki and Ohshima et al. disclosed a novel catalytic method for the direct synthesis of aliphatic diazenes from sterically hindered α -tertiary amines. The authors demonstrate the broad substrate applicability of this method, its potential in hetero-diazene synthesis, and its successful application in Fmoc solid-phase peptide synthesis. The experimental section presents comprehensive optimization studies and mechanistic investigations, supporting the validity of the proposed catalytic system. Given the importance of deaminative functionalization in synthetic organic chemistry, particularly for C-N bond transformations, this study is highly relevant. Overall, I suggest a major revision before publication.

- (1) The reaction time only takes a few minutes. What is the reason for such high efficiency?
- (2) The proposed catalytic cycle relies on DFT calculations without experimental validation. Key intermediates (e.g., copper(III) amide species B) are not detected or characterized. Control experiments (e.g., radical trapping, EPR studies) to confirm the aminyl radical pathway are absent. The authors should address these issues.
- (3) In Table 1, why are the yields much lower when using DCDMH and DIDMH compared to when using DBDMH?
- (4) In Table 1, Entry 12, "NBS (3.0 equiv) of DBDMH" should be replaced with "NBS (3.0 equiv) instead of DBDMH."
- (5) In Table 2, when the authors investigated hetero-diazene synthesis, there were some changes in the reaction conditions. How do these conditions prevent the formation of self-coupling products?
- (6) α -Secondary amines tend to form undesirable imines, so the authors primarily focused on the diazene synthesis from α -tertiary amines. I'm curious, under the standard conditions, can aryl amines, such as aniline, give the desired product?
- (7) When using a substrate containing two α -tertiary amines, such as 2,5-dimethylhexane-2,5-diamine, is it possible to obtain the intramolecular cyclization product 3,3,6,6-tetramethyl-3,4,5,6-tetrahydropyridazine?

Reviewer #2

(Remarks to the Author)

Yazaki et al. presented a catalytic method for synthesis of azo compounds from α -tertiary amines for deaminative functionalization.

Study is supported by extensive optimization studies with solvent, catalysts, base and oxidant. DFT calculations have been included for understanding the mechanistic pathway.

They investigated the substrate scope involving hetero-coupling with two distinct α -tertiary amines.

The methodology's applicability was demonstrated with peptide-derived diazene synthesis. The repetitive sequence peptide of elastin was synthesized through Fmoc solid-phase peptide synthesis.

The direct transformation of α -tertiary amine into tertiary alkyl halides was achieved using the presented methodology.

Under blue light irradiation, hetero-diazenes constructed highly congested contiguous quaternary carbon centers, which are generally challenging to assemble, through solvent-cage directed radical cross-coupling.

The presence of a copper catalyst, essential for the azo bond formation, was demonstrated through control experiments.

The presented catalyzed method enabled the transformation of C-N bonds into C-halogen, C-H, C-O, C-S, C-Se, and C-C (Csp³, Csp², Csp) bonds through a deamination carbon-centered radical formation utilizing diazenes with a broad substrate scope.

Considering the novelty of the presented strategy, I support its publication in Nature Communications.

Minor corrections: Some compounds such as 3py, 3cp, 20 requires purification and spectra may be included in S.I.

Some phrase may be rewritten such as "Katritzky salts cannot be synthesized from

sterically hindered α -tertiary amines, however, which limits their applicability" for better understanding.

Reviewer #3

(Remarks to the Author)

The authors described a Cu-catalyzed synthesis of diazenes from α -tertiary amine and their synthetic application. As (correctly) pointed out by the authors, this work succeeds in filling a substantial gap in the synthesis of sterically hindered diazenes via a catalytic manner. The scope and usefulness were extensively demonstrated with elaborated substrates, especially, the hetero-coupling examples, albeit with compromised atom-economy and efficiency. Synthetic application was also demonstrated in solid state peptide synthesis and a panel of product transformations. These results highlight the usefulness of sterically hindered diazene as free radical precursors and in the preparation of useful molecules with sterically hindered sp³-sp³ moieties, which would be attractive to synthetic and medicinal chemists. The manuscript is well organized with adequate citation of literature precedents, and the product characterization and synthetic procedures have also been well-presented. Mechanistic insights were also obtained with a series of control experiments and DFT studies (although somewhat flawed, see below).

In view of the above rationale, I think this manuscript could be considered for publication on Nature Commun., with major revisions as detailed below.

1. This protocol seems to work efficiently for homo- and hetero-coupling with tertiary amines, while falls short in the reactions between a tertiary and a secondary amine (2x-2z), or between two secondary amines (3py, 13% yield), although higher loadings of Cu salt were used. A rationale for this difference of reactivity should be commented, as it relates directly to the mechanistic proposal and offer potential applications by the community. Would be a Cu-amide species undergoing beta-H elimination in operation? What about the barriers of oxidative addition, halogen atom abstraction events for these substrates? (see comments on the DFT part)

2. The readers might be curious about the relative thermal stability of sterically hindered diazine products for different reasons. Some experimental data for the thermal stabilities of representative compounds would be highly useful (such as TGA analysis or decomposition profile in a high boiling point solvent).

3. The authors proposed a mechanism based on detailed control experiments, NMR studies and DFT studies. While the control experiments and NMR studies provide highly useful insights, the DFT studies are somewhat flawed in a few aspects. (1) M06-2X behaves well for main group elements but not for transition metals. With Cu being considered here, M06-2X is not expected to give convincing results, and M06 would be much better in doing this job. While I understand that the authors may have desired better results in the Cu-free steps (such as Figure (2)), a re-check of the validity of the results in Figure (1-1) should be performed with a more suitable functional. For all atoms 6-311+G(d,p) basis set was used in single point calculation, which is not recommended, as relativistic effect should be considered for Br. In addition, the exact form of the functional, such as r- or u- form, should be specified. The unrestricted open (u-) form should be applied in this case (i.e., ub3lyp and uM06-2X). As the default methods for open-shell and closed-shell species are different in Gaussian, simply using the key words "b3lyp" or "M06-2X" would lead to errors, as it treats Cu(I) and Cu(III) complexes with r-form while the Cu(II) and radical species with u-form. The authors should check this point carefully. A generally important, but (maybe) less-significant issue here is the consideration of dispersion correction, as b3lyp is extremely poor in describing this effect. (2) In section 9-2, (1-1), a detailed thermodynamic loop for the interaction of amide bromide is provided, which is very useful for the mechanistic understanding. A graphical energy profile is suggested to be added for the readers to better comprehend the results more distinctively. In addition, according to the authors calculation, direct oxidative addition to Cu(III) is feasible, but the barriers of the halogen-atom abstraction (XAA) steps are not given. The readers would have no clue whether the XAA steps are more favored.

(3) For the N-N bond formation step, the authors calculated the possible combinations of the two reacting species and based on kinetics and thermodynamics of the reaction, proposed that (ii)-TS would be a more likely pathway. The data per se would sound valid. However, in comparison with the experimental conditions (1 min at 5 mol% Cu loading!), a barrier of 27.4 kcal/mol would be a bit too high. Another possible pathway according to this reviewer is the radical-radical recombination, which would be highly kinetically favorable, and would also better explain the statistic distribution of the products in the cross-coupling examples with two distinct amines in the scope studies. Evidences or calculations for or against such a possibility should be provided.

Version 1:

Reviewer comments:

Reviewer #1

(Remarks to the Author)

The authors have adequately addressed the previous concerns and improved the clarity of the key findings. The revised results now robustly support their conclusions, and I believe the work is scientifically rigorous and suitable for publication in Nature Communications.

Reviewer #2

(Remarks to the Author)

The authors have revised their manuscript according to the suggestions from the reviewers, I therefore recommend acceptance of this manuscript for publication.

Reviewer #3

(Remarks to the Author)

The revised manuscript has been significantly improved, especially the DFT part. Now the calculated energy profile agrees well with the experimental results and provides a good rationale. I am enthusiastic on the publication of this manuscript on Nature Commun. in the current form.

One minor issue, which may not influence the integrity of the current work but the authors should pay attention to in future works: that in my previous comments, I meant the authors should use the "u" form across the board, regardless of the spin multiplicity, because both closed shell and open shell species were involved. Now that in the revised manuscript good agreement was observed and the trend of the reactivity has not changed, it might suggest that the errors in this case are small and negligible. Such treatment should nonetheless be followed for more rigorous results and to avoid re-doing the whole calculation in other works.

REVIEWER COMMENTS

Reviewer #1 (Remarks to the Author):

In this manuscript, Yazaki and Ohshima et al. disclosed a novel catalytic method for the direct synthesis of aliphatic diazenes from sterically hindered α -tertiary amines. The authors demonstrate the broad substrate applicability of this method, its potential in hetero-diazene synthesis, and its successful application in Fmoc solid-phase peptide synthesis. The experimental section presents comprehensive optimization studies and mechanistic investigations, supporting the validity of the proposed catalytic system. Given the importance of deaminative functionalization in synthetic organic chemistry, particularly for C-N bond transformations, this study is highly relevant. Overall, I suggest a major revision before publication.

- (1) The reaction time only takes a few minutes. What is the reason for such high efficiency?
- (2) The proposed catalytic cycle relies on DFT calculations without experimental validation. Key intermediates (e.g., copper(III) amide species B) are not detected or characterized. Control experiments (e.g., radical trapping, EPR studies) to confirm the aminyl radical pathway are absent. The authors should address these issues.
- (3) In Table 1, why are the yields much lower when using DCDMH and DIDMH compared to when using DBDMH?
- (4) In Table 1, Entry 12, "NBS (3.0 equiv) of DBDMH" should be replaced with "NBS (3.0 equiv) instead of DBDMH."
- (5) In Table 2, when the authors investigated hetero-diazene synthesis, there were some changes in the reaction conditions. How do these conditions prevent the formation of self-coupling products?
- (6) α -Secondary amines tend to form undesirable imines, so the authors primarily focused on the diazene synthesis from α -tertiary amines. I'm curious, under the standard conditions, can aryl amines, such as aniline, give the desired product?
- (7) When using a substrate containing two α -tertiary amines, such as 2,5-dimethylhexane-2,5-diamine, is it possible to obtain the intramolecular cyclization product 3,3,6,6-tetramethyl-3,4,5,6-tetrahydropyridazine?

Reviewer #2 (Remarks to the Author):

Yazaki et al. presented a catalytic method for synthesis of azo compounds from α -tertiary amines for deaminative functionalization.

Study is supported by extensive optimization studies with solvent, catalysts, base and oxidant. DFT calculations have been included for understanding the mechanistic pathway.

They investigated the substrate scope involving hetero-coupling with two distinct α -tertiary amines.

The methodology's applicability was demonstrated with peptide-derived diazene synthesis. The repetitive sequence peptide of elastin was synthesized through Fmoc solid-phase peptide synthesis.

The direct transformation of α -tertiary amine into tertiary alkyl halides was achieved using the presented methodology.

Under blue light irradiation, hetero-diazenes constructed highly congested contiguous quaternary carbon centers, which are generally challenging to assemble, through solvent-cage directed radical cross-coupling.

The presence of a copper catalyst, essential for the azo bond formation, was demonstrated through control experiments.

The presented catalyzed method enabled the transformation of C–N bonds into C–halogen, C–H, C–O, C–S, C–Se, and C–C (Csp³, Csp², Csp) bonds through a deamination carbon-centered radical formation utilizing diazenes with a broad substrate scope.

Considering the novelty of the presented strategy, I support its publication in Nature Communications.

Minor corrections: Some compounds such as 3py, 3cp, 20 requires purification and spectra may be included in S.I.

Some phrase may be rewritten such as "Katritzky salts cannot be synthesized from sterically hindered α -tertiary amines, however, which limits their applicability" for better understanding.

Reviewer #3 (Remarks to the Author):

The authors described a Cu-catalyzed synthesis of diazenes from α -tertiary amine and their synthetic application. As (correctly) pointed out by the authors, this work succeeds in filling a substantial gap in the synthesis of sterically hindered diazenes via a catalytic manner. The scope and usefulness were extensively demonstrated with elaborated substrates, especially, the hetero-coupling examples, albeit with compromised atom-economy and efficiency. Synthetic application was also demonstrated in solid state peptide synthesis and a panel of product transformations. These results highlight the usefulness of sterically hindered diazene as free radical precursors and in the preparation of useful molecules with sterically hindered sp³-sp³ moieties, which would be attractive to synthetic and medicinal chemists. The manuscript is well organized with adequate citation of literature precedents, and the product characterization and synthetic procedures have also been well-presented. Mechanistic insights were also obtained with a series of control experiments and DFT studies (although somewhat flawed, see below).

In view of the above rationale, I think this manuscript could be considered for publication on Nature Commun., with major revisions as detailed below.

1. This protocol seems to work efficiently for homo- and hetero-coupling with tertiary amines, while falls short in the reactions between a tertiary and a secondary amine (2x-2z), or between two secondary amines (3py, 13% yield), although higher loadings of Cu salt were used. A rationale for this difference of reactivity should be commented, as it relates directly to the mechanistic proposal and offer potential applications by the community. Would be a Cu-amide species undergoing beta-H elimination in operation? What about the barriers of oxidative addition, halogen atom abstraction events for these substrates? (see comments on the DFT part)
2. The readers might be curious about the relative thermal stability of sterically hindered diazine products for different reasons. Some experimental data for the thermal stabilities of

representative compounds would be highly useful (such as TGA analysis or decomposition profile in a high boiling point solvent).

3. The authors proposed a mechanism based on detailed control experiments, NMR studies and DFT studies. While the control experiments and NMR studies provide highly useful insights, the DFT studies are somewhat flawed in a few aspects.

(1) M06-2X behaves well for main group elements but not for transition metals. With Cu being considered here, M06-2X is not expected to give convincing results, and M06 would be much better in doing this job. While I understand that the authors may have desired better results in the Cu-free steps (such as Figure (2)), a re-check of the validity of the results in Figure (1-1) should be performed with a more suitable functional. For all atoms 6-311+G(d,p) basis set was used in single point calculation, which is not recommended, as relativistic effect should be considered for Br. In addition, the exact form of the functional, such as r- or u-form, should be specified. The unrestricted open (u-) form should be applied in this case (i.e., ub3lyp and uM06-2X). As the default methods for open-shell and closed-shell species are different in Gaussian, simply using the key words "b3lyp" or "M06-2X" would lead to errors, as it treats Cu(I) and Cu(III) complexes with r-form while the Cu(II) and radical species with u-form. The authors should check this point carefully. A generally important, but (maybe) less-significant issue here is the consideration of dispersion correction, as b3lyp is extremely poor in describing this effect.

(2) In section 9-2, (1-1), a detailed thermodynamic loop for the interaction of amide bromide is provided, which is very useful for the mechanistic understanding. A graphical energy profile is suggested to be added for the readers to better comprehend the results more distinctively. In addition, according to the authors calculation, direct oxidative addition to Cu(III) is feasible, but the barriers of the halogen-atom abstraction (XAA) steps are not given. The readers would have no clue whether the XAA steps are more favored.

(3) For the N-N bond formation step, the authors calculated the possible combinations of the two reacting species and based on kinetics and thermodynamics of the reaction, proposed that (ii)-TS would be a more likely pathway. The data per se would sound valid. However, in comparison with the experimental conditions (1 min at 5 mol% Cu loading!), a barrier of 27.4 kcal/mol would be a bit too high. Another possible pathway according to this reviewer is the radical-radical recombination, which would be highly kinetically favorable, and would also better explain the statistic distribution of the products in the cross-coupling examples with two distinct amines in the scope studies. Evidences or calculations for or against such a possibility should be provided.

Reviewer #1 (Remarks to the Author):

Reviewer Comment 1

The reaction time only takes a few minutes. What is the reason for such high efficiency?

Answer

We thank you for your invaluable comments. Based on the reviewer's comments, we recalculated the reaction pathway for the N–N bond formation. As a result, the ΔG^\ddagger value decreased from 27.4 to 8.8 kcal/mol. This value is consistent with the experimental observation that the reaction proceeds rapidly at room temperature. Furthermore, the proposed aminyl radical intermediate was detected as a DMPO adduct, indicating the generation of a highly reactive nitrogen radical. We believe this high reactivity facilitates efficient N–N bond formation, which accounts for the observed rapid reaction.

Reviewer Comment 2

The proposed catalytic cycle relies on DFT calculations without experimental validation. Key intermediates (e.g., copper(III) amide species B) are not detected or characterized. Control experiments (e.g., radical trapping, EPR studies) to confirm the aminyl radical pathway are absent. The authors should address these issues.

Answer

We sincerely appreciate the reviewer's constructive comments. In accordance with the reviewer's suggestions, we conducted additional experiments to trap the aminyl radical. To precisely control the amount of oxidant, we employed NBS instead of DBDMH, and successfully detected the DMPO adduct by HRMS and EPR. The EPR signal was found to be in good agreement with the simulated spectrum.

Regarding the Cu(III) species, we recalculated the pathway as instructed by the reviewer. The pathway involving oxidative addition to form a Cu(III) species followed by aminyl radical generation was found to have a the ΔG value of 25.8 kcal/mol. Based on these results, we conclude that the oxidative addition pathway is unlikely to be the dominant mechanism. Instead, the halogen abstraction pathway appears to be more plausible.

Reviewer Comment 3

In Table 1, why are the yields much lower when using DCDMH and DIDMH compared to when using DBDMH?

Answer

We thank you for your invaluable comments. When DCDMH was used, the *N*-chlorinated intermediate was found to be stable. This intermediate was observed as the major product, suggesting that the formation of the active aminyl radical is relatively slow under these conditions. In contrast, when DIDMH was used, no *N*-iodinated species were detected, and the reaction system became more complex. *N*-diiodoamines have been reported to be highly unstable, forming polymeric mixtures including R-NH₂, and are therefore considered unsuitable for use under the current reaction conditions (*Angew. Chem. Int. Ed.* **1962**, *1*, 46). The detailed screening results are provided in Section 8-2 of the SI.

Reviewer Comment 4

In Table 1, Entry 12, "NBS (3.0 equiv) of DBDMH" should be replaced with "NBS (3.0 equiv) instead of DBDMH.

Answer

We thank you for your invaluable comments. We revised according to the reviewer's comment.

Entry	Variation from standard conditions	Yield (%)
1	None	>99
2	No CuOAc	7
3	Cu(OAc) ₂ instead of CuOAc	22
4	Fe(OAc) ₂ instead of CuOAc	21
5	AgOAc instead of CuOAc	35
6	O ₂ (1 atm) instead of DBDMH	0
7	Oxone (3.0 equiv) instead of DBDMH	0
8	K ₂ S ₂ O ₈ (3.0 equiv) instead of DBDMH	0
9	TBHP (3.0 equiv) instead of DBDMH	0
10	DCDMH instead of DBDMH	33
11	DIDMH instead of DBDMH	4
12	NBS (3.0 equiv) instead of DBDMH	>99
13	K ₂ CO ₃ instead of DBU	30
14	KO ^t Bu instead of DBU	25
15	NEt ₃ instead of DBU	5
16	DABCO instead of DBU	60

Reactions were conditions on a 0.20 mmol scale.
Yields were determined by ¹H NMR spectroscopic analysis.

Reviewer Comment 5

In Table 2, when the authors investigated hetero-diazeno synthesis, there were some changes in the reaction conditions. How do these conditions prevent the formation of self-coupling products?

Answer

We sincerely appreciate the reviewer's constructive comments. As detailed in the Supplementary Information, it was in fact difficult to suppress self-coupling. Through optimization, we found that the most critical factor for improving the yield of the cross-coupled product was the use of an excess (3 equivalents) of one of the substrates. Assuming a random coupling scenario, the theoretical yield of the cross-coupled product is 75% when 3 equivalents are used, and 67% when 2 equivalents are used. The experimental yields were 71% and 66%, respectively, which are in good agreement with the theoretical values.

Reviewer Comment 6

α -Secondary amines tend to form undesirable imines, so the authors primarily focused on the diazene synthesis from α -tertiary amines. I'm curious, under the standard conditions, can aryl amines, such as aniline, give the desired product?

Answer

We thank you for your invaluable comments. When the reaction was performed under the optimized conditions using aniline, the NMR yield was 18%, and a large amount of brominated byproducts through Friedel-Crafts reaction was observed. To date, diazene formation from aniline has been reported under halogen-free conditions, such as with a copper catalyst, pyridine, and an oxygen atmosphere (*Angew. Chem. Int. Ed.* **2010**, *49*, 6174). These results suggest that conditions avoiding the use of halogenating agents are more suitable for aniline substrates.

Reviewer Comment 7

When using a substrate containing two α -tertiary amines, such as 2,5-dimethylhexane-2,5-diamine, is it possible to obtain the intramolecular cyclization product 3,3,6,6-tetramethyl-3,4,5,6-tetrahydropyridazine?

Answer

Thank you very much for your valuable advice. We examined the intramolecular diazene formation using 2,5-dimethylhexane-2,5-diamine, but found that the starting material was consumed, resulting in a complex mixture. Although we performed DART-MS analysis, the desired product was not detected under the current optimized conditions. Based on the reviewer's suggestions, we intend to continue exploring intramolecular reactions in future studies. Thank you again for your helpful feedback.

Reviewer #2 (Remarks to the Author):

Reviewer Comment 1

Minor corrections: Some compounds such as 3py, 3cp, 20 requires purification and spectra may be included in S.I.

Answer

We thank you for your invaluable comments. For compound 3py, we attempted further purification. However, we were unable to completely eliminate trace impurities. Therefore, we have revised the reported yield to "< 13%" and added a note indicating the presence of minor impurities. For compounds 3cp (in CD₃OD) and 20 (in acetone-d₆), we repeated the reaction and purification, and replaced the NMR spectra with newly obtained data. As a result, the yield of 3cp was updated from 54% to 51%.

Benzyl 1-((E)-((1s,3s)-adamantan-1-yl)diazene)cyclopentane-1-carboxylate (3cp): (Changed condition : 1 h., Colorless oil, *n*-Hexane/Et₂O = 100/1 to 50/1, 51% yield, 37.0 mg); ¹H NMR (500 MHz, CD₃OD) δ 7.25–7.17 (m, 5H, ArH), 5.01 (s, 2H, OCH₂), 2.11–2.00 (m, 7H, AdH, CH₂CH₂), 1.68–1.66 (m, 3H, AdH, CH₂CH₂), 1.63–1.53 (m, 13H, AdH, CH₂CH₂); ¹³C NMR (125 MHz, CD₃OD) δ 173.1, 136.2, 128.0, 127.8, 127.7, 85.7, 67.3, 66.1, 39.8, 36.2, 33.7, 29.2, 24.3; IR (neat) 2936, 2903, 2851, 1736, 1452, 1261, 1165, 1069, 1016, 733, 694 cm⁻¹; HRMS (DART) *m/z* calc'd. for C₂₃H₃₁N₂O₂ (M + H)⁺ 367.2380, found 367.2368.

Benzyl 1-((1s,3s)-adamantan-1-yl)cyclohexane-1-carboxylate (20): (Changed condition : 0.060 mmol, White solid, *n*-Hexane/Et₂O = 100/1 to 50/1 to 10/1 to 5/1, 42% yield, 9.0 mg); ¹H NMR (500 MHz, Acetone-d₆) δ 7.46–7.44 (m, 2H, ArH), 7.40–7.37 (m, 2H, ArH), 7.35–7.31 (m, 1H, ArH), 5.16 (s, 2H, ArCH₂), 2.13–2.11 (m, 2H, CH₂CH₂CH₂), 1.92 (br, 3H, AdH), 1.66–1.52 (m, 15H, AdH, CH₂CH₂CH₂), 1.29–1.18 (m, 4H, CH₂CH₂CH₂), 1.13–1.05 (m, 1H, CH₂CH₂CH₂); ¹³C NMR (125 MHz, Acetone-d₆) δ 174.3, 137.1, 129.0, 128.8, 128.3, 65.8, 54.6, 37.5, 37.4, 37.2, 29.0, 28.0, 26.0, 24.5; IR (neat) 2941, 2905, 1707, 1447, 1260, 1209, 1175, 1125, 1082, 1024, 802, 750, 694, 604 cm⁻¹; HRMS (DART) *m/z* calc'd. for C₂₄H₃₃O₂ (M + H)⁺ 353.2475, found 353.2462.

Reactions were conducted on 0.20 mmol scale using 0.2 ml of DMF. Isolated yields are shown. Diastereomeric ratios were determined by ¹H NMR analysis of crude mixture.

^a 1-HCl salt (1.0 equiv), DBU (2.0 equiv). ^b 1-TFA salt (1.0 equiv), DBU (2.0 equiv). ^c DMF (0.4 ml), 3 h. ^d DCDMH was used instead of DBDMH. ^e CuOAc (20 mol%). ^f CuOAc (50 mol%).

^g CuOAc (100 mol%), 1.0 mmol scale. ^h Three equivalents of the amine on the left side were used. 0.4 ml of DMF. ⁱ Reaction time was 1 h. ^j 0.50 mmol scale. ^k CuOAc (100 mol%). ^l A small amount of impurity was still present after silica gel column chromatography. Boc, *tert*-butoxycarbonyl; Cbz, benzyloxycarbonyl; TBDPS, *tert*-butyldiphenylsilyl; TBS, *tert*-butyldimethylsilyl.

We also added the following comment to the main text.

Cross-coupling of α -tertiary amine with α -secondary amine successfully provided the hetero-diazene **3py**, although the chemical yield was modest and a small amount of impurity remained.

Reviewer Comment 2

Some phrase may be rewritten such as "Katritzky salts cannot be synthesized from sterically hindered α -tertiary amines, however, which limits their applicability" for better understanding.

Answer

We thank you for your invaluable comments. The sentence you pointed out has been corrected as follows.

However, Katritzky salts cannot be synthesized from sterically hindered α -tertiary amines, which limits their applicability.

Reviewer #3 (Remarks to the Author):

Reviewer Comment 1

This protocol seems to work efficiently for homo- and hetero-coupling with tertiary amines, while falls short in the reactions between a tertiary and a secondary amine (2x-2z), or between two secondary amines (3py, 13% yield), although higher loadings of Cu salt were used. A rationale for this difference of reactivity should be commented, as it relates directly to the mechanistic proposal and offer potential applications by the community. Would be a Cu-amide species undergoing beta-H elimination in operation? What about the barriers of oxidative addition, halogen atom abstraction events for these substrates? (see comments on the DFT part) may be included in S.I.

Answer

We sincerely appreciate the reviewer's constructive comments. In response to the reviewer's constructive comment, we performed additional experiments using α -secondary amines to investigate the effect of the amount of copper catalyst on imine formation (the scheme was added in SI). The results showed that increasing the amount of copper catalyst suppressed the formation of the imine; for example, the imine yield decreased from 57% (with 5 mol% Cu catalyst) to approximately 20% when 100 mol% of the catalyst was used. We also examined the effect of using weaker bases; however, these conditions did not prevent imine formation. These findings indicate that while the copper catalyst can suppress the formation of the imine to some extent, imine formation is not completely avoided even when stoichiometric amounts of copper are used, suggesting that copper-mediated pathways may still contribute to imine formation.

In light of the reviewer's suggestion, we also re-evaluated the reaction mechanism via DFT calculations. The updated computational analysis supports the hypothesis that aminyl radical formation is more likely to proceed via a halogen abstraction pathway rather than oxidative addition. However, the calculations also suggest that although oxidative addition itself is energetically feasible, the subsequent generation of the aminyl radical from the resulting copper(III) species is energetically unfavorable. Therefore, we speculate that β -hydride elimination from the copper(III) intermediate is more likely to occur under these conditions.

We also added the following comment to the main text.

Although increasing the amount of copper catalyst effectively suppressed the formation of the imine byproduct, the observation of imine formation even under stoichiometric copper conditions indicates that copper-mediated pathways may still contribute to imine formation.

Reviewer Comment 2

The readers might be curious about the relative thermal stability of sterically hindered diazine products for different reasons. Some experimental data for the thermal stabilities of representative compounds would be highly useful (such as TGA analysis or decomposition profile in a high boiling point solvent).

Answer

We thank you for your invaluable comments. We performed TGA for some representative compounds, according to the advice. The data are added in the revised SI (Figure S1). The results demonstrated that the thermal stability of these compounds was comparable to, or significantly greater than, that of a typical diazene compound (i.e., azobisisobutyronitrile, AIBN). These mentions are added in the revised manuscript.

We also added the following comment to the main text.

We investigated thermal stability using thermogravimetric analysis (TGA) for several representative compounds, namely 2a, 2r, 2t, 2y, and 3ap, which possess different hindered structures (Figure S1). The thermal stability of these compounds was comparable to, or significantly greater than, that of a typical diazene compound (i.e., azobisisobutyronitrile, AIBN), which shows an onset of thermal decomposition at approximately 100–125 °C⁵³. Therefore, the synthesis reported here may enhance the availability of thermally stable diazene compounds, which could also be valuable in the field of polymer science, as these compounds are commonly used for radical polymerization.

Reviewer Comment 3

The authors proposed a mechanism based on detailed control experiments, NMR studies and DFT studies. While the control experiments and NMR studies provide highly useful insights, the DFT studies are somewhat flawed in a few aspects.

(1) M06-2X behaves well for main group elements but not for transition metals. With Cu being considered here, M06-2X is not expected to give convincing results, and M06 would be much better in doing this job. While I understand that the authors may have desired better results in the Cu-free steps (such as Figure (2)), a re-check of the validity of the results in Figure (1-1) should be performed with a more suitable functional. For all atoms 6-311+G(d,p) basis set was used in single point calculation, which is not recommended, as relativistic effect should be considered for Br. In addition, the exact form of the functional, such as r- or u-form, should be specified. The unrestricted open (u-) form should be applied in this case (i.e., ub3lyp and uM06-2X). As the default methods for open-shell and closed-shell species are different in Gaussian, simply using the key words "b3lyp" or "M06-2X" would lead to errors, as it treats Cu(I) and Cu(III) complexes with r-form while the Cu(II) and radical species with u-form. The authors should check this point carefully. A generally important, but (maybe) less-significant issue here is the consideration of dispersion correction, as b3lyp is extremely poor in describing this effect.

Answer

We sincerely thank the reviewer for the insightful and technically detailed feedback regarding our DFT calculations. In accordance with the reviewer's suggestions, we have re-evaluated the proposed reaction pathway. For bromine-containing systems, geometry optimizations were performed using the LANL2DZ basis set with the corresponding effective core potential (ECP), while single-point energy calculations were conducted using the SDD basis set along with its associated ECP. Additionally, the functional used for single-point calculations was changed to M06 in consideration of the involvement of copper atoms. The revised computational results have been reflected in the figures in the Supporting Information.

The choice between restricted and unrestricted formalisms was carefully made based on the open-shell or closed-shell nature of each species, and this information is now explicitly provided in the Supporting Information.

As the reviewer noted, B3LYP is not ideal for dispersion-sensitive systems. However, test calculations with dispersion correction showed negligible differences from these without dispersion correction, suggesting that dispersion effects are not significant in this case. Therefore, dispersion corrections were omitted in the geometry optimizations.

Reviewer Comment 4

(2) In section 9-2, (1-1), a detailed thermodynamic loop for the interaction of amide bromide is provided, which is very useful for the mechanistic understanding. A graphical energy profile is suggested to be added for the readers to better comprehend the results more distinctively. In addition, according to the authors calculation, direct oxidative addition to Cu(III) is feasible, but the barriers of the halogen-atom abstraction (XAA) steps are not given. The readers would have no clue whether the XAA steps are more favored.

Answer

As suggested by the reviewer, graphical energy profiles have been added to the Supporting Information to facilitate understanding of the proposed reaction pathway.

Transition-state calculations involving open-shell systems with heavy atoms such as copper and bromine are widely recognized as highly challenging. We are currently working on the transition-state calculations for aminyl radical formation; however, due to the computational limitations, we do not expect to obtain reliable results within the time frame of the peer review. We plan to present these results, along with a detailed discussion, in a separate publication.

At this stage, based on the thermodynamic energy differences, the halogen abstraction pathway appears to be more likely than the oxidative addition pathway. Preliminary transition-state calculations also indicate that the oxidative addition intermediate can reversibly convert to a precursor of the halogen abstraction process. We have updated the manuscript and the Supporting Information accordingly to reflect these findings.

Reviewer Comment 5

(3) For the N-N bond formation step, the authors calculated the possible combinations of the two reacting species and based on kinetics and thermodynamics of the reaction, proposed that (ii)-TS would be a more likely pathway. The data per ser would sound valid. However, in comparison with the experimental conditions (1 min at 5 mol% Cu loading!), a barrier of 27.4 kcal/mol would be a bit too high. Another possible pathway according to this reviewer is the radical-radical recombination, which would be highly kinetically favorable, and would also better explain the statistic distribution of the products in the cross-coupling examples with two distinct amines in the scope studies. Evidences or calculations for or against such a possibility should be provided.

Answer

Based on DFT calculations using appropriate functionals and basis sets, we found that the activation barrier for the N-N bond formation step is sufficiently low to enable the reaction to proceed rapidly at room temperature (8.8 kcal/mol for (ii)-TS). A comparison of the different pathways showed that the relative trends in both activation energies and thermodynamic energy differences were consistent with those obtained in our original calculations. Although the transition-state structure for pathway (iv) could not be converged, this pathway can be excluded on thermodynamic grounds. Pathway (ii) is the most favorable both kinetically and thermodynamically, as supported by spin-trapping experiments detecting the N-H aminyl radical; however, pathways (i) and (iii) cannot be completely excluded. We plan to report a more detailed mechanistic study and calculations in a future publication.

Under the catalytic conditions, the aminyl radical is expected to exist only at very low concentrations, making homocoupling between aminyl radicals unlikely. While the radical-radical coupling is almost barrier-less, the feasibility of this pathway is inherently dependent

on the radical concentration. As such, its relevance cannot be accurately assessed by computational analysis alone, and we consider this pathway to be mechanistically disfavored under the present reaction conditions.

sp UM06/SDD(Br)-6-311+G(d,p)/SMD(DMF)//UB3LYP/Lan12dz(Br)-6-31+G(d,p) at 1 atm, 298.15 K.
 The values of Gibbs free energy are given in kcal/mol.

REVIEWER COMMENTS

Reviewer #1 (Remarks to the Author):

The authors have adequately addressed the previous concerns and improved the clarity of the key findings. The revised results now robustly support their conclusions, and I believe the work is scientifically rigorous and suitable for publication in Nature Communications.

Reviewer #2 (Remarks to the Author):

The authors have revised their manuscript according to the suggestions from the reviewers, I therefore recommend acceptance of this manuscript for publication.

Reviewer #3 (Remarks to the Author):

The revised manuscript has been significantly improved, especially the DFT part. Now the calculated energy profile agrees well with the experimental results and provides a good rationale. I am enthusiastic on the publication of this manuscript on Nature Commun. in the current form.

One minor issue, which may not influence the integrity of the current work but the authors should pay attention to in future works: that in my previous comments, I meant the authors should use the "u" form across the board, regardless of the spin multiplicity, because both closed shell and open shell species were involved. Now that in the revised manuscript good agreement was observed and the trend of the reactivity has not changed, it might suggest that the errors in this case are small and negligible. Such treatment should nonetheless be followed for more rigorous results and to avoid re-doing the whole calculation in other works.

Reviewer #3 (Remarks to the Author):

Reviewer Comment 1

One minor issue, which may not influence the integrity of the current work but the authors should pay attention to in future works: that in my previous comments, I meant the authors should use the "u" form across the board, regardless of the spin multiplicity, because both closed shell and open shell species were involved. Now that in the revised manuscript good agreement was observed and the trend of the reactivity has not changed, it might suggest that the errors in this case are small and negligible. Such treatment should nonetheless be followed for more rigorous results and to avoid re-doing the whole calculation in other works.

Answer

We sincerely thank the reviewer for the positive evaluation of our revised manuscript. Regarding Reviewer #3's helpful remark on the use of unrestricted ("u") calculations across all species, we greatly appreciate this valuable guidance. In response, we recalculated several intermediates and transition states using the unrestricted ("u") form in place of the restricted ("r") form, and found that the energy differences arising specifically from the choice between these two forms were less than 0.2 cal/mol (0.0002 kcal/mol). Based on this result, we concluded that such differences are negligible in the context of our current study. Nevertheless, we will be sure to adopt this recommendation in future investigations to further enhance the reliability and reproducibility of our computational analyses. Once again, we thank the reviewer for their insightful and constructive feedback, which has greatly improved the quality and rigor of our work.